# SSFL: Tackling Label Deficiency in Federated Learning via Personalized Self-Supervision

## Abstract

Federated Learning (FL) is transforming the ML training ecosystem from a centralized over-the-cloud setting to distributed training over edge devices in order to strengthen data privacy, reduce data migration costs, and break regulatory restrictions. An essential, but rarely studied, challenge in FL is *label deficiency* at the edge. This problem is even more pronounced in FL, compared to centralized training, due to the fact that FL users are often reluctant to label their private data and edge devices do not provide an ideal interface to assist with annotation. Addressing label deficiency is also further complicated in FL, due to the heterogeneous nature of the data at edge devices and the need for developing personalized models for each user. We propose a self-supervised and personalized federated learning framework, named (`SSFL`), and a series of algorithms under this framework which work towards addressing these challenges. First, under the `SSFL` framework, we analyze the compatibility of various centralized self-supervised learning methods in FL setting and demonstrate that `SimSiam` networks performs the best with the standard `FedAvg` algorithm. Moreover, to address the data heterogeneity at the edge devices in this framework, we have innovated a series of algorithms that broaden existing supervised personalization algorithms into the setting of self-supervised learning including `perFedAvg`, `Ditto`, and local fine-tuning, among others. We further propose a novel personalized federated self-supervised learning algorithm, `Per-SSFL`, which balances personalization and consensus by carefully regulating the distance between the local and global representations of data. To provide a comprehensive comparative analysis of all proposed algorithms, we also develop a distributed training system and related evaluation protocol for `SSFL`. Using this training system, we conduct experiments on a synthetic non-I.I.D. dataset based on CIFAR-10, and an intrinsically non-I.I.D. dataset GLD-23K. Our findings show that the gap of evaluation accuracy between supervised learning and unsupervised learning in FL is both small and reasonable. The performance comparison indicates that representation regularization-based personalization method is able to outperform other variants. Ablation studies on SSFL are also conducted to understand the role of batch size, non-I.I.D.ness, and the evaluation protocol.

## 1 Introduction

Federated Learning (FL) is a contemporary distributed machine learning paradigm that aims at strengthening data privacy, reducing data migration costs, and breaking regulatory restrictions (Kairouz et al., 2021; Wang et al., 2021). It has been widely applied to computer vision, natural language processing, and data mining. However, there are two main challenges impeding its wider adoption in machine learning. One is data heterogeneity, which is a natural property of FL in which diverse clients may generate datasets with different distributions due to behavior preferences (e.g., the most common cause of heterogeneity is skewed label distribution which might result from instances where some smartphone users take more landscape pictures, while others take more photos of daily life). The second challenge is label deficiency at the edge, which is relatively less studied. This issue is more severe at the edge than in a centralized setting because users are reluctant to annotate their private and sensitive data, and/or smartphones and IoT devices do not have a user-friendly interface to assist with annotation.

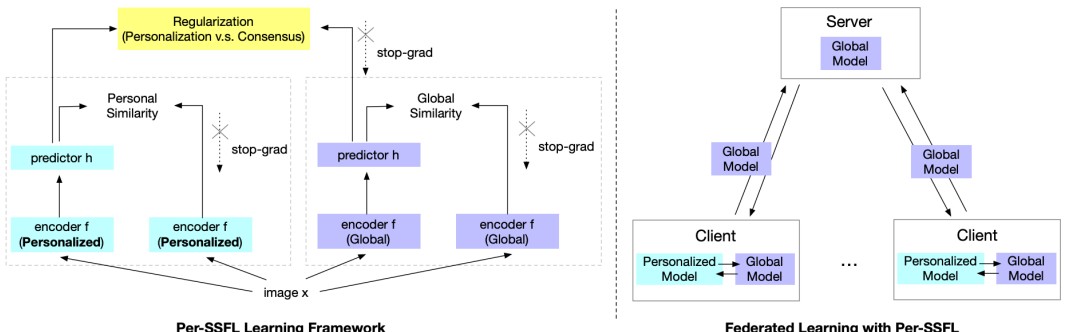

Figure 1: Depiction of the Self-supervised and Personalized Federated Learning (SSFL) framework.

To mitigate the data heterogeneity issue among clients, researchers have proposed algorithms for training a global model `FedAvg` (McMahan et al., 2017), `FedProx` (Li et al., 2018), `FedNova` (Wang et al., 2020), `FedOPT` (Reddi et al., 2020), as well as personalized FL frameworks (e.g., `pFedMe`, `Ditto`, `Per-FedAvg`). These algorithms all depend on the strong assumption that the data at the edge has sufficient labels. To address the label deficiency issue in FL, recent works (Liu et al., 2020; Long et al., 2020; Itahara et al., 2020; Jeong et al., 2020; Liang et al., 2021; Zhao et al., 2020; Zhang et al., 2020a;b) assume that the server or client has a fraction of labeled data and use semi-supervised methods such as consistency loss (Miyato et al., 2018) or pseudo labeling (Lee, 2013) to train a global model. A more realistic but challenging setting is fully unsupervised training. Although a recent work in FL (Saeed et al., 2020) attempts to address this challenge through Siamese networks proposed around thirty years ago (Bromley et al., 1993), its design does not tackle data heterogeneity for learning personalized models, and it only trains on small-scale sensor data in IoT devices. Moreover, these existing works in FL have not examined recent progress in the Self-Supervised Learning (SSL) community where methods such as `SimCLR` (Chen et al., 2020), `SwAV`(Caron et al., 2021), `BYOL` (Grill et al., 2020), and `SimSiam` (Chen & He, 2020) have shown tremendous improvement in reducing the amount of labeled data required to achieve state-of-the-art performance. As such, it remains still unclear how these SSL methods can be incorporated into FL and how well they would perform, especially when intertwined with the data heterogeneity challenge that does not exist in centralized training.

In this paper, we propose Self-Supervised Federated Learning (`SSFL`), a unified self-supervised and personalized federated learning framework, and a series of algorithms under this framework to address these challenges. As shown in Figure 1, this framework brings state-of-the-art self-supervised learning algorithms to the realm of FL in order to enable training without using any supervision, while also integrating model personalization to deal with data heterogeneity (Section 3.1). More specifically, under the `SSFL` framework, we analyze the compatibility of various centralized self-supervised learning methods in the FL setting and demonstrate that `SimSiam` networks performs the best with the standard FedAvg algorithm (Section 3.2). Moreover, to address the data heterogeneity at edge devices, we have innovated a series of algorithms that broaden the reach of existing supervised personalization algorithms into the setting of self-supervised learning, including `perFedAvg` (Fallah et al., 2020), `Ditto` (Li et al., 2021), and `local fine-tuning`, among others. We further propose a novel personalized federated self-supervised learning algorithm, `per-SSFL` (Section 3.3), which balances personalization and consensus by carefully regulating the distance between the local and global representations of data (shown as the yellow block in Figure 1).

To provide a comprehensive and comparative analysis of the proposed algorithms, we also develop a distributed training system and evaluation protocol for SSFL. Using this training system, we conduct experiments on a synthetic non-I.I.D. dataset based on CIFAR-10 and a natural non-I.I.D. dataset GLD-23K. Our experimental results demonstrate that all algorithms in our framework work reliably. In FL, the gap of evaluation accuracy between supervised learning and unsupervised learning is small. Personalized `SSFL` performs better than `FedAvg`-based `SSFL`. We also conduct ablation studies to fully understand the `SSFL` framework, namely the role of batch size, different degrees of non-I.I.D.ness, and performance in more datasets. Finally, our unified API design can serve as a suitable platform and baseline, enabling further developments of more advanced `SSFL` algorithms.

## 2 Preliminaries

SSFL builds upon two fundamental areas in machine learning: federated optimization and self-supervised learning. Thus, we first introduce some basics and formulations in these areas.

### 2.1 Federated Optimization

Federated optimization refers to the distributed optimization paradigm that a network of $K$ devices collaboratively solve a machine learning task. In general, it can be formulated as a distributed optimization problem with the form (McMahan et al., 2017): $\min_\theta \sum_{k=1}^K \frac{|D_k|}{|D|} \mathcal{L}(\theta, D_k)$. Here, each device $k$ has a local dataset $D_k$ drawn from a local distribution $X_k$. The combined dataset $D = \cup_{k=1}^K D_k$ is the union of all local datasets $D_k$. $\theta$ represents the model weight of a client model. $\mathcal{L}$ is the client's local loss function that measures the local empirical risk over the heterogeneous dataset $\mathcal{D}^k$. Under this formulation, to mitigate the non-I.I.D. issue, researchers have proposed algorithms such as FedProx (Li et al., 2018), FedNova (Wang et al., 2020), and FedOPT (Reddi et al., 2020) for training a global model, as well as personalized FL frameworks such as Ditto (Li et al., 2021), and Per-FedAvg (Fallah et al.). All of these algorithms have a strong assumption that data at the edge have sufficient labels, meaning that their analysis and experimental settings are based on a supervised loss function, such as the cross-entropy loss for image classification.

### 2.2 Self-supervised Learning

Self-supervised learning (SSL) aims to learn meaningful representations of samples without human supervision. Formally, it aims to learn an encoder function $f_\theta : \mathcal{X} \mapsto \mathbb{R}^d$ where $\theta$ is the parameter of the function, $\mathcal{X}$ is the unlabeled sample space (e.g. image, text), and the output is a $d$ dimensional vector containing enough information for downstream tasks such as image classification and segmentation. The key to SSL's recent success is the inductive bias that ensures a good representation encoder remains consistent under different perturbations of the input (i.e. consistency regularization).

One prominent example among recent advances in modern SSL frameworks is the Siamese architecture (Bromley et al., 1993) and its improved variants SimCLR (Chen et al., 2020), SwAV (Caron et al., 2021), BYOL (Grill et al., 2020), and SimSiam (Chen & He, 2020). Here we review the most elegant architecture, SimSiam, and defer the description and comparison of the other three to Appendix A. SimSiam proposes a two-head architecture in which two different views (augmentations) of the same image are encoded by the same network $f_\theta$. Subsequently, a predictor Multi Layer Perceptron (MLP) $h_\theta$ and a *stop-gradient* operation denoted by $\widehat{\cdot}$ are applied to both heads. In the SSL context, "stop gradient" means that the optimizer stops at a specific neural network layer during the back propagation and views the parameters in preceding layers as constants. Here, $\theta$ is the concatenation of the parameters of the encoder network and the predictor MLP. The algorithm aims to minimize the negative cosine similarity $\mathcal{D}(\cdot, \cdot)$ between two heads. More concretely, the loss is defined as

$$\mathcal{L}_{\mathrm{SS}}(\theta, D) = \frac{1}{|D|} \sum_{x \in D} \mathcal{D}(f_\theta(\mathcal{T}(x)), \widehat{h_\theta(f_\theta(\mathcal{T}(x)))}), \tag{1}$$

where $\mathcal{T}$ represents stochastic data augmentation and $D$ is the data set.

## 3 SSFL: Self-supervised Federated Learning

In this section, we propose SSFL, a unified framework for self-supervised federated learning. Specifically, we introduce the method by which SSFL works for collaborative training of a global model and personalized models, respectively.

### 3.1 General Formulation

We formulate self-supervised federated learning as the following distributed optimization problem:

$$\min_{\substack{\Theta \\ \{\theta_k\}_{k \in [K]}}} G\left(\mathcal{L}\left(\theta_1, \Theta; X_1\right), \ldots, \mathcal{L}\left(\theta_K, \Theta; X_K\right)\right) \tag{2}$$

where $\theta_k$ is the parameter for the local model $(f_{\theta_k}, h_{\theta_k})$; $\Theta$ is the parameter for the global model $(f_\Theta, h_\Theta)$; $\mathcal{L}(\theta_k, \Theta; X_k)$ is a loss measuring the quality of representations encoded by $f_{\theta_k}$ and $f_\Theta$ on the local distribution $X_k$; and $G(\cdot)$ denotes the aggregation function (e.g. sum of client losses weighted by $\frac{|D_k|}{|D|}$). To capture the two key challenges in federated learning (data heterogeneity and label deficiency), we hold two core assumptions in the proposed framework: (1) $X_k$ of all clients are heterogeneous (non-I.I.D.), and (2) there is no label.

To tackle the above problem, we propose a unified training framework for federated self-supervised learning, as described in Algorithm 1. This framework can handle both non-personalized and personalized federated training. In particular, if one enforces the constraint $\theta_k = \Theta$ for all clients $k \in [K]$, the problem reduces to learning a global model. When this constraint is not enforced, $\theta_k$ can be different for each client, allowing for model personalization. ClientSSLOPT is the local optimizer at the client side which solves the local sub-problem in a self-supervised manner. ServerOPT takes the update from the client side and generates a new global model for all clients.

---

**Algorithm 1:** SSFL: A Unified Framework for Self-supervised Federated Learning

**input** : $K, T, \Theta^{(0)}, \{\theta_k^{(0)}\}_{k \in [K]}$, CLIENTSSLOPT , SERVEROPT

1   **for** $t = 0, \ldots, T-1$ **do**
2      Server randomly selects a subset of devices $S^{(t)}$
3      Server sends the current global model $\Theta^{(t)}$ to $S^{(t)}$
4      **for** *device* $k \in S^{(t)}$ *in parallel* **do**
5          Solve local sub-problem of equation 2:

$$\theta_k, \Theta_k^{(t)} \leftarrow \text{CLIENTSSLOPT}\,(\theta_k^{(t)}, \Theta^{(t)}, \nabla\mathcal{L}(\theta_k, \Theta; X_k))$$

6          Send $\Delta_k^{(t)} := \Theta_k^{(t)} - \Theta^{(t)}$ back to server
7      $\Theta^{(t+1)} \leftarrow \text{SERVEROPT}\,\left(\Theta^{(t)}, \{\Delta_k^{(t)}\}_{k \in S^{(t)}}\right)$

**return** : $\{\theta_k\}_{k \in [K]}, \Theta^{(T)}$

---

Next, we will introduce specific forms of ClientSSLOPT and ServerOPT for global training and personalized training.

## 3.2   GLOBAL-SSFL: COLLABORATION TOWARDS A GLOBAL MODEL WITHOUT SUPERVISION

To train a global model using SSFL, we design a specific form of ClientSSLOPT using SimSiam. We choose SimSiam over other contemporary self-supervised learning frameworks (e.g., SimSiam, SwAV, BYOL) based on the following analysis as well as experimental results (see Section 5.1).

**The simplicity in neural architecture and training method.** SimSiam's architecture and training method are relatively simple. For instance, compared with SimCLR, SimSiam has a simpler loss function; compared with SwAV, SimSiam does not require an additional neural component (prototype vectors) and Sinkhorn-Knopp algorithm; compared with BYOL, SimSiam does not need to maintain an additional moving averaging network for an online network. Moreover, the required batch size of SimSiam is the smallest, making it relatively friendly for resource-constrained federated learning. A more comprehensive comparison can be found in Appendix A.

**Interpretability of SimSiam leads to simpler local optimization.** More importantly, SimSiam is more interpretable from an optimization standpoint which simplifies the local optimization. In particular, it can be viewed as an implementation of an Expectation-Maximization (EM) like algorithm, meaning that optimizing $\mathcal{L}_{\text{SS}}$ in Equation 1 is implicitly optimizing the following objective

$$\min_{\theta, \eta} \mathbb{E}_{\underset{x \sim X}{\mathcal{T}}} \left[ \|f_\theta(\mathcal{T}(x)) - \eta_x\|_2^2 \right]. \tag{3}$$

Here, $f_\theta$ is the encoder neural network parameterized by $\theta$. $\eta$ is an extra set of parameters, whose size is proportional to the number of images, and $\eta_x$ refers to using the image index of $x$ to access a sub-vector of $\eta$. This formulation is w.r.t. both $\theta$ and $\eta$ and can be optimized via an alternating algorithm. At time step $t$, the $\eta_x^t$ update takes the form $\eta_x^t \leftarrow \mathbb{E}_{\mathcal{T}}[f_{\theta^t}(\mathcal{T}(x))]$, indicating that $\eta_x^t$ is assigned the average representation of $x$ over the distribution of augmentation. However, it is impossible to compute this step by going over the entire dataset during training. Thus, SimSiam uses one-step optimization to approximate the EM-like two-step iteration by introducing the predictor $h_\theta$

to approximate $\eta$ and learn the expectation (i.e. $h_\theta(z) \approx \mathbb{E}_\mathcal{T}[f_\theta(\mathcal{T}(x))]$) for any image $x$. After this approximation, the expectation $\mathbb{E}_\mathcal{T}[\cdot]$ is ignored because the sampling of $\mathcal{T}$ is implicitly distributed across multiple epochs. Finally, we can obtain the self-supervised loss function in Equation 1, in which negative cosine similarity $\mathcal{D}$ is used in practice (the equivalent $L_2$ distance is used in Equation 3 for the sake of analysis). Applying equation 1 as `ClientSSLOPT` simplifies the local optimization for each client in a self-supervised manner.

### 3.3 PER-SSFL: LEARNING PERSONALIZED MODELS WITHOUT SUPERVISION

In this section, we explain how SSFL addresses the data heterogeneity challenge by learning personalized models. Inspired by the interpretation in Section 3.2, we define the following sub-problem for each client $k \in [K]$:

$$\min_{\theta_k, \eta_k} \quad \mathbb{E}_{\substack{x \sim X_k \\ \mathcal{T}}} \left[ \|f_{\theta_k}(\mathcal{T}(x)) - \eta_{k,x}\|_2^2 + \frac{\lambda}{2} \|\eta_{k,x} - \mathcal{H}_x^*\|_2^2 \right]$$

$$\text{s.t.} \quad \Theta^*, \mathcal{H}^* \in \arg\min_{\Theta, \mathcal{H}} \sum_{i=1}^n \frac{|D_k|}{|D|} \mathbb{E}_{\substack{x \sim X_i \\ \mathcal{T}}} \left[ \|f_\Theta(\mathcal{T}(x)) - \mathcal{H}_x\|_2^2 \right] \tag{4}$$

Compared to global training, we additionally include $\Theta$, the global model parameter, and $\mathcal{H}$, the global version of $\eta$, and the expected representations which correspond to the personalized parameters $\theta_k$ and $\eta_k$. In particular, through the term $\|\eta_{k,x} - \mathcal{H}_x^*\|_2^2$, we aim for the expected local representation of any image $x$ to reside within a neighborhood around the expected global representation of $x$. Therefore, by controlling the radius of the neighborhood, hyperparameter $\lambda$ helps to balance consensus and personalization.

---

**Algorithm 2:** Per-SSFL

**input :** $K, T, \lambda, \Theta^{(0)}, \{\theta_i^{(0)}\}_{k \in [K]}, s$: number of local iteration, $\beta$: learning rate

1 **for** $t = 0, \ldots, T-1$ **do**
2    Server randomly selects a subset of devices $S^{(t)}$
3    Server sends the current global model $\Theta^{(t)}$ to $S^{(t)}$
4    **for** *device $k \in S^{(t)}$ in parallel* **do**
5      Sample mini-batch $B_k$ from local dataset $D_k$, and do $s$ local iterations
     /* Optimize the global parameter $\Theta$ locally          */
6      $Z_1, Z_2 \leftarrow f_{\Theta^{(t)}}(\mathcal{T}(B_k)), f_{\Theta^{(t)}}(\mathcal{T}(B_k))$
7      $P_1, P_2 \leftarrow h_{\Theta^{(t)}}(Z_1), h_{\Theta^{(t)}}(Z_2)$
8      $\Theta_k^{(t)} \leftarrow \Theta^{(t)} - \beta\nabla_{\Theta^{(t)}} \frac{\mathcal{D}(P_1, \widehat{Z_2}) + \mathcal{D}(P_2, \widehat{Z_1})}{2}$, where $\widehat{\cdot}$ stands for stop-gradient
     /* Optimize the local parameter $\theta_k$          */
9      $z_1, z_2 \leftarrow f_{\theta_k}(\mathcal{T}(B_k)), f_{\theta_k}(\mathcal{T}(B_k))$
10      $p_1, p_2 \leftarrow h_{\theta_k}(z_1), h_{\theta_k}(z_2)$
11      $\theta_k \leftarrow \theta_k - \beta\nabla_{\theta_k} \left( \frac{\mathcal{D}(p_1, \widehat{z_2}) + \mathcal{D}(p_2, \widehat{z_1})}{2} + \lambda \frac{\mathcal{D}(p_1, P_1) + \mathcal{D}(p_1, P_2) + \mathcal{D}(p_2, P_1) + \mathcal{D}(p_2, P_2)}{4} \right)$    ClientSSLOpt
12      Send $\Delta_k^{(t)} := \Theta_k^{(t)} - \Theta^{(t)}$ back to server
13    $\Theta^{(t+1)} \leftarrow \Theta^{(t)} + \sum_{k \in S^{(t)}} \frac{|D_k|}{|D|} \Delta_k^{(t)}$    SERVEROPT

**return :** $\{\theta_i\}_{i \in [n]}, \Theta^{(T)}$

---

We see that Equation 4 in the above objective is an optimization problem w.r.t. both $\theta$ and $\eta$. However, as the above target is intractable in practice, following an analysis similar to Section 3.2, we use the target below as a surrogate:

$$\min_{\theta_k} \quad \mathcal{L}_{\text{SS}}(\theta_k, D_k) + \frac{\lambda}{|D_k|} \sum_{x \in D_k} \mathcal{D}\left(h_{\theta_k}(f_{\theta_k}(\mathcal{T}(x))), h_{\Theta^*}(f_{\Theta^*}(\mathcal{T}(x)))\right) \tag{5}$$

$$\text{s.t.} \quad \Theta^* \in \arg\min_\Theta \mathcal{L}_{\text{SS}}(\Theta, D) \tag{6}$$

In practice, $\Theta$ can be optimized independently of $\theta^k$ through the `FedAvg` (McMahan et al., 2017) algorithm. To make the computation more efficient, we also apply the symmetrization trick proposed in (Chen & He, 2020). We refer to this algorithm as `Per-SSFL` and provide a detailed description in Algorithm 2 (also illustrated in Fig. 1).

**Regarding the theoretical analysis.** To our knowledge, all self-supervised learning frameworks do not have any theoretical analysis yet, particularly the SimSiam dual neural network architecture.

Our formulation and optimization framework are interpretable, they are built based on an EM-like algorithm for SimSiam and minimizing the distance between the private model and the global model's data representation.

**Innovating baselines to verify SSFL.** *Note that we have not found any related works that explore a Siamese-like SSL architecture in an FL setting.* As such, to investigate the performance of our proposed algorithm, we further propose several other algorithms that can leverage the `SSFL` framework. 1. `LA-SSFL`. We apply `FedAvg` (McMahan et al., 2017) on the `SimSiam` loss $\mathcal{L}_{SS}$ for each client to obtain a global model. We perform one step of SGD on the clients' local data for local adaption; 2. `MAML-SSFL`. This algorithm is inspired by perFedAvg (Fallah et al.) and views the personalization on each devices as the inner loop of MAML (Finn et al.). It aims to learn an encoder that can be easily adapted to the clients' local distribution. During inference, we perform one step of SGD on the global model for personalization; 3. `BiLevel-SSFL`. Inspired by Ditto (Li et al., 2021), we learn personalized encoders on each client by restricting the parameters of all personalized encoders to be close to a global encoder independently learned by weighted aggregation. More details of these algorithms, formulation, and pseudo code are introduced in Appendix B. In Section 5.3, we will show the comparison results for these proposed `SSFL` algorithmic variants.

## 4 TRAINING SYSTEM AND EVALUATION PIPELINE FOR SSFL

**A Distributed Training System to accelerate the algorithmic exploration in SSFL framework.** We also contributed to reproducible research via our distributed training system. This is necessary for two reasons: (1) Running a stand-alone simulation (training client by client sequentially) like most existing FL works requires a prohibitively long training time when training a large number of clients. In SSFL, the *model size* (e.g., ResNet-18 v.s. shallow CNNs used in the original FedAvg paper) and the *round number* for convergence (e.g., 800 epochs in the centralized SimSiam framework) is relatively larger than in FL literature. By running all clients in parallel on multiple CPUs/GPUs, we can largely accelerate the process. (2) Given that SSFL is a unified and generic learning framework, researchers may develop more advanced ways to improve our work. As such, we believe it is necessary to design unified APIs and system frameworks in line with the algorithmic aspect of SSFL. See Appendix C for more details on our distributed training system.

**Evaluation Pipeline.** In the training phase, we use a `KNN` classifier (Wu et al., 2018) as an online indicator to monitor the quality of the representations generated by the `SimSiam` encoder. For `Global-SSFL`, we report the `KNN` test accuracy using the global model and the global test data, while in `Per-SSFL`, we evaluate all clients' local encoders separately with their local test data and report their averaged accuracy. After self-supervised training, to evaluate the performance of the trained encoder, we freeze the encoder and attach a linear classifier to the output of the encoder. For `Global-SSFL`, we can easily verify the performance of `SimSiam` encoder by training the attached linear classifier with `FedAvg`. However, for `Per-SSFL`, each client learns a personalized `SimSiam` encoder. As the representations encoded by personalized encoders might reside in different spaces, using a single linear classifier trained by `FedAvg` to evaluate these representations is unreasonable (see experiments in Section 5.4.3). As such, we suggest an additional evaluation step to provide a more representative evaluation of `Per-SSFL`'s performance: for each personalization encoder, we use the entire training data to train the linear classifier but evaluate on each client's local test data.

## 5 EXPERIMENTS

In this section, we introduce experimental results for SSFL with and without personalization and present a performance analysis on a wide range of aspects including the role of batch size, different degrees of non-IIDness, and understanding the evaluation protocol.

**Implementation.** We develop the SSFL training system to simplify and unify the algorithmic exploration. Details of the training system design can be found in Appendix C. We deploy the system in a distributed computing environment which has 8 NVIDIA A100-SXM4 GPUs with sufficient memory (40 GB/GPU) to explore different batch sizes (Section 5.4.1). Our training framework can run multiple parallel training workers in a single GPU, so it supports federated training with a large number of clients. The client number selected per round used in all experiments is 10, which is a reasonable setting suggested by recent literature (Reddi et al., 2020).

**Learning Task.** Following `SimCLR` (Chen et al., 2020), `SimSiam` (Chen & He, 2020), `BYOL` (Grill et al., 2020), and `SwAV` (Caron et al., 2020) in the centralized setting, we evaluate SSL for the *image classification* task and use representative datasets for federated learning.

**Dataset.** We run experiments on synthetic non-I.I.D. dataset CIFAR-10 and intrinsically non-I.I.D. dataset Google Landmark-23K (GLD-23K), which are suggested by multiple canonical works in the FL community (Reddi et al., 2020; He et al., 2020; Kairouz et al., 2019). For the non-I.I.D. setting, we distribute the dataset using a Dirichlet distribution (Hsu et al., 2019), which samples $\mathbf{p}_c \sim \mathrm{Dir}(\alpha)$ (we assume a uniform prior distribution) and allocates a $\mathbf{p}_{c,k}$ proportion of the training samples of class $c$ to local client $k$. We provide a visualization of the data distribution in Appendix E.1.

**Model Architecture.** For the model architecture, ResNet-18 is used as the backbone of the `SimSiam` framework, and the predictor is the same as that in the original paper.

Next, we focus on results from the curated CIFAR-10 dataset and defer GLD-23K to Appendix D.

## 5.1 COMPARISONS ON SIMSIAM, SIMCLR, SWAV, AND BYOL

Our first experiment determines which SSL method is ideal for FL settings. We run experiments using FedAvg for these four methods and obtain two findings: (1) `SimSiam` outperforms `SimCLR` in terms of accuracy; (2) `BYOL` and `SwAV` do not work in FL; we tested a wide range of hyper-parameters, but they still are unable to converge to a reasonable accuracy. These experimental results confirm our analysis in Section 3.2.

## 5.2 EVALUATION ON GLOBAL-SSFL

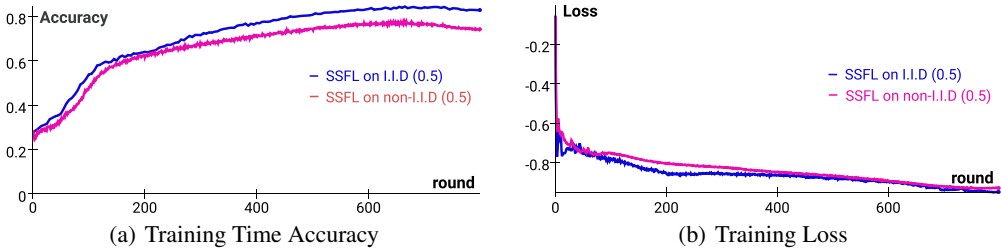

(a) Training Time Accuracy        (b) Training Loss

Figure 2: Training and Evaluation using SSFL

The goal of this experiment is to understand the accuracy gap between supervised and self-supervised federated learning in both I.I.D. and non-I.I.D. settings where we aim to train a global model from private data from clients.

**Setup and Hyper-parameters.** We evaluate Global-SSFL using non-I.I.D. data from CIFAR-10: we set $\alpha = 0.1$ for the Dirichlet distribution. For supervised learning, the test accuracy is evaluated on a global test dataset. For self-supervised training, we follow the evaluation protocol introduced in Section 4. We use SGD with Momentum as the client-side optimizer and a learning rate scheduler across communication rounds. We searched the learning rate on the grid of $\{0.1, 0.3, 0.01, 0.03\}$ and report the best result. The experiment is run three times using the same learning rate with fixed random seeds to ensure reproducibility. The training lasts for 800 rounds, which is sufficient to achieve convergence for all methods. More hyperparameters are in Appendix E.2.

We display the training curves in Figure 10 which demonstrates that `SSFL` can converge reliably in both I.I.D. and non-I.I.D. settings. For the I.I.D. data, we find that `SSFL` can achieve the same accuracy as the centralized accuracy report in the `SimSiam` paper (Chen & He, 2020). For the non-I.I.D. data, SSFL achieves a reasonable accuracy compared to the centralized accuracy. The accuracy comparisons in different dimensions (supervised v.s. self-supervised; I.I.D. v.s. non-I.I.D.) are summarized in Table 1.

|  | Accuracy | | |
|---|---|---|---|
|  | Supervised | Self-Supervised | Acc. Gap |
| I.I.D | 0.932 | 0.914 | 0.018 |
| non-I.I.D | 0.8812 | 0.847 | 0.0342 |
| Acc. Gap | 0.0508 | 0.06 | N/A |

Table 1: Evaluation accuracy comparison between supervised FL and `SSFL`.

| Method | KNN Indicator | Evaluation |
|---|---|---|
| `LA-SSFL` | 0.9217 | 0.8013 |
| `MAML-SSFL` | 0.9355 | 0.8235 |
| `BiLevel-SSFL` | 0.9304 | 0.8137 |
| `Per-SSFL` | 0.9388 | 0.8310 |

Table 2: Evaluation Accuracy for Various Per-SSFL Methods.

## 5.3 EVALUATION ON PER-SSFL

Based on the results of SSFL with `FedAvg`, we further add the personalization components for SSFL introduced in Section 3.3 (Per-SSFL).

**Setup and Hyper-parameters.** For a fair comparison, we evaluate `Per-SSFL` on non-I.I.D. data from CIFAR-10 and set $\alpha = 0.1$ for the Dirichlet distribution. For `Per-SSFL` training, we follow the evaluation protocol introduced in Section 4. Similar to SSFL, we use SGD with Momentum as the client-side optimizer and the learning rate scheduler across communication rounds. We search for the learning rate on a grid of $\{0.1, 0.3, 0.01, 0.03\}$ and report the best result. For `Per-SSFL` and `BiLevel-SSFL`, we also tune the $\lambda$ of the regularization term with a search space $\{1, 10, 0.1, 0.01, 0.001\}$. The experiments are run three times with the same learning rate and with fixed random seeds to ensure reproducibility. The training also lasts for 800 communication rounds, which is the same as `Global-SSFL`. Other hyperparameters can be found in Appendix E.2.

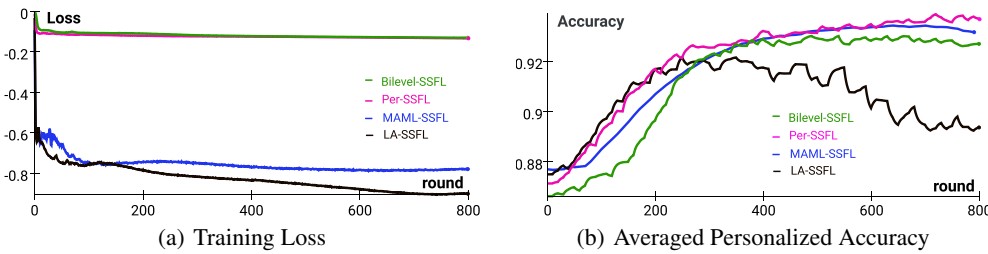

(a) Training Loss  (b) Averaged Personalized Accuracy

Figure 3: Training and Evaluation using SSFL

We illustrate our results in Figure 10 and Table 2. To confirm the convergence, we draw loss curves for all methods in Figure 10(b) (note that they have different scaled values due to the difference of their loss functions). Figure 10(b) indicates that `Per-SSFL` performs best among all methods. `MAML-SSFL` is also a suggested method since it obtains comparable accuracy. `LA-SSFL` is a practical method, but it does not perform well in the self-supervised setting. In Figure 10(b), the averaged personalized accuracy of `LA-SSFL` diverges in the latter phase. Based on `BiLevel-SSFL`'s result, we can conclude such a method is not a strong candidate for personalization, though it shares similar objective functions as `Per-SSFL`. This indicates that regularization through representations encoded by `SimSiam` outperforms regularization through weights.

## 5.4 PERFORMANCE ANALYSIS

### 5.4.1 ROLE OF BATCH SIZE

FL typically requires a small batch size to enable practical training on resource-constrained edge devices. Therefore, understanding the role of batch size in `SSFL` is essential to practical deployment and applicability. To investigate this, we use different batch sizes and tune the learning rate to find the optimal accuracy for each one. The results in Figure 4 show that SSFL requires a large batch size (256); otherwise, it reduces the accuracy or diverges during training. Since system efficiency is not the focus of this paper, we use gradient accumulation, which is a simple yet effective method. We fix the batch size at 32 and use accumulation step 8 for all experiments. For an even larger batch size (e.g., 512), the memory cost is significant, though there is no notable gain in accuracy. Therefore, we discontinue the search for batch sizes larger than larger than 256. A more advanced method includes batch-size-one training and knowledge distillation. We defer the discussion to Appendix F.

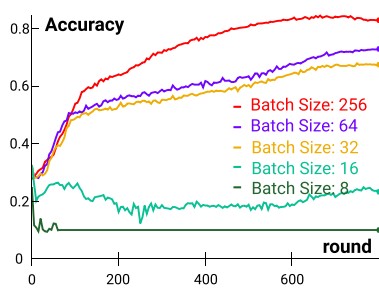

Figure 4: Results for batch sizes

### 5.4.2 ON DIFFERENT DEGREES OF NON-I.I.D.NESS

We investigate the impact of the degree of data heterogeneity on the SSFL performance. We compare the performance between $\alpha = 0.1$ and $\alpha = 0.5$. These two settings provide a non-negligible gap

in the label distribution in each client (see our visualization in Appendix E.1). Figure 5(a) and 5(b) shows the learning curve comparisons. It is clearly observed that the higher degree of data heterogeneity makes it converge more slowly, adversely affecting the accuracy.

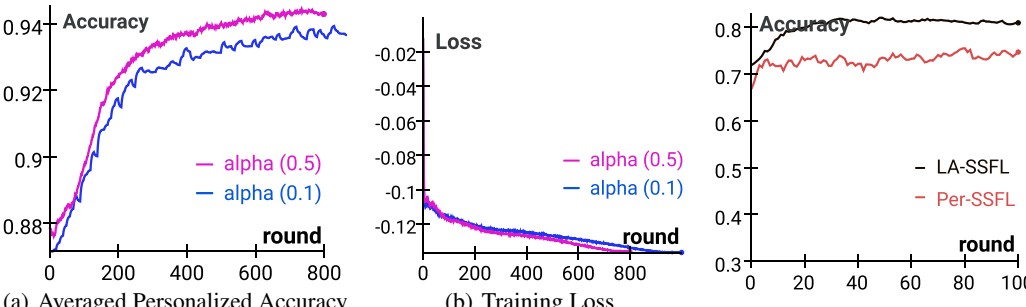

(a) Averaged Personalized Accuracy    (b) Training Loss

Figure 5: Evaluation on Different Degress of Non-I.I.D.ness

Figure 6: Understanding the Evaluation Protocol

### 5.4.3 UNDERSTANDING THE LINEAR EVALUATION OF PERSONALIZED ENCODERS

As we discussed in 4, in SSFL, we can easily verify the quality of the SimSiam encoder using federated linear evaluation; however, in Per-SSFL, each client learns a personalized SimSiam encoder. Such heterogeneity in diverse encoders makes a fair evaluation difficult. To demonstrate this, we run experiments with naive federated linear evaluation on personalized encoders and surprisingly find that such an evaluation protocol downgrades the performance. As shown in Figure 6, the federated linear evaluation for Per-SSFL performs worse than even LA-SSFL. This may be attributed to the fact that the naive aggregation drags close to the parameter space of all heterogeneous encoders, making the encoder degenerate in terms of personalization.

## 6 RELATED WORKS

**Federated Learning (FL) with Personalization.** pFedMe (Dinh et al.), perFedAvg (Fallah et al., 2020), and Ditto (Li et al., 2021) are some representative works in this direction. However, these methods all have a strong assumption that users can provide reliable annotations for their private and sensitive data, which we argue to be very unrealistic and impractical.

**Label deficiency in FL.** There are a few related works to tackle label deficiency in FL (Liu et al., 2020; Long et al., 2020; Itahara et al., 2020; Jeong et al., 2020; Liang et al., 2021; Zhao et al., 2020; Zhang et al., 2020b). Compared to these works, our proposed SSFL does not use any labels during training. FedMatch (Jeong et al., 2020) and FedCA (Zhang et al., 2020a) requires additional communication costs to synchronize helper models or public labeled dataset. (Saeed et al., 2020) addresses the fully unsupervised challenge on small-scale sensor data in IoT devices. However, compared to our work, it uses the Siamese networks proposed around thirty years ago (Bromley et al., 1993), lacking consideration on the advance in the past two years (i.e., SimCLR (Chen et al., 2020), SwAV(Caron et al., 2021), BYOL (Grill et al., 2020), and SimSiam (Chen & He, 2020)). Moreover, these works does not have any design for learning personalized models.

## 7 CONCLUSION

We propose Self-supervised Federated Learning (SSFL) framework and a series of algorithms under this framework towards addressing two challenges: data heterogeneity and label deficiency. SSFL can work for both global model training and personalized model training. We conduct experiments on a synthetic non-I.I.D. dataset based on CIFAR-10 and the intrinsically non-I.I.D. GLD-23K dataset. Our experimental results demonstrate that SSFL can work reliably and achieves reasonable evaluation accuracy that is suitable for use in various applications.

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
