# OpenReview forum: "SSFL: Tackling Label Deficiency in Federated Learning via Personalized Self-Supervision"
_ICLR.cc/2022/Conference — ICLR 2022 Submitted_

### Official Review · Reviewer_zbEP · 2021-11-02

**Correctness:** 3
**Technical Novelty And Significance:** 3
**Empirical Novelty And Significance:** 2
**Recommendation:** 3
**Confidence:** 3

**Main Review:**


Strengths
- Deals with an important and practical scenario of FL
- The analysis of personalised representations is interesting

Weakness
- Insufficient quantitative assessment of SimSiam approach against other self-supervised approaches.
- It is unclear the feature representation can be used in practice, especially in
the personalised setting. Perhaps explore the use as an auxiliary objective in some downstream applications?
- The effect of hyperparameters is under-explored. For instance, the size of representation which could overpower the benefits of self-supervision in the assessment with KNN.


**Summary Of The Paper:**

This work is concerned with a very practical scenario of Federated Learning where the participating agents may not have access to labelled data. SimSiam architecture to learn useful feature representations with extensions that incorporate personalisation for local client models. The learnt representation are evaluated against a KNN classifier for analysing their usefulness.


**Summary Of The Review:**

I think the work tackles an important question, however the experimentation is inadequate to assess the usefulness of the approach.

---

> ### Author Response · Authors · 2021-11-23
> **Author Response**
>
> * Insufficient quantitative assessment of SimSiam approach against other self-supervised approaches.
>
> Response: "Insufficiency" is such a general vocabulary. Can you provide specific suggestions? In which aspects, how should it be evaluated?
>
> * It is unclear the feature representation can be used in practice, especially in the personalised setting. Perhaps explore the use as an auxiliary objective in some downstream applications?
>
> Response: I believe our experimental results already demonstrate that personalized method can improve the accuracy. Normally, as many SSL related works in centralized setting, if the performance of SSL pretraining is improved, the downstream performance is also improved. As in FL, it's a proof of a concept, given that there is a lack of downstream evaluation dataset for FL, it's hard for us to do this evaluation. But we indeed appreciate your suggestion. Thanks.
>
> * The effect of hyperparameters is under-explored. For instance, the size of representation which could overpower the benefits of self-supervision in the assessment with KNN.
>
> Response: This is reasonable, but from SSL literature in centralized setting, the size of representation is not explored. I think it's unfair to ask FL to evaluate it.
>
> ---
> In addition, we believe the following contributions build a solid baseline for research in this direction, and our open-source code may further enhance this endeavor.
>
> (Global-SSFL) A self-supervised federated learning framework for a global model.
>
> Global-SSFL’s interpretability in optimization perspective.
>
> (Per-SSFL) A novel self-supervised federated learning framework for personalized models. The optimization method is novel, compared to the simple combination of Per-FL and SimSiam. Note that the optimizer we proposed is based on our large-scale experimental exploration, it's not that straightforward.
>
> (Many algorithmic variants of Per-SSFL) We also proposed many variants of Per-SSFL that does not perform better than our novel formulation and algorithm. These algorithms serve as the baseline for our novel Per-SSFL.
>
> A new evaluation metric for Per-SSFL. Please check Section 3.
>
> A real distributed training system makes this research possible (evaluating different algorithmic variants in a reasonable time; exploring the memory constraint led by large batch size training on SSL, etc.). It’s also very important for future research in this area.
>
> ---
>
> We need more detailed and technical comments. Please reconsider your rating. Thanks.

---

### Official Review · Reviewer_bsjD · 2021-11-02

**Correctness:** 1
**Technical Novelty And Significance:** 1
**Empirical Novelty And Significance:** Not applicable
**Recommendation:** 1
**Confidence:** 5

**Main Review:**

See Summary Of The Paper for detail.

**Summary Of The Paper:**

The submission is not anonymized, the author's name chaoyanghe appears frequently in the attached SSFL_Sumpplementary/SSFL-Source-Code. Therefore, it should be directly desk rejected.

--- Logging error ---
Traceback (most recent call last):
  File "/Users/chaoyanghe/opt/anaconda3/envs/fedml/lib/python3.7/logging/__init__.py", line 1025, in emit
    msg = self.format(record)
  File "/Users/chaoyanghe/opt/anaconda3/envs/fedml/lib/python3.7/logging/__init__.py", line 869, in format
    return fmt.format(record)
  File "/Users/chaoyanghe/opt/anaconda3/envs/fedml/lib/python3.7/logging/__init__.py", line 608, in format
    record.message = record.getMessage()
  File "/Users/chaoyanghe/opt/anaconda3/envs/fedml/lib/python3.7/logging/__init__.py", line 369, in getMessage
    msg = msg % self.args

**Summary Of The Review:**

See Summary Of The Paper for detail.

---

> ### Author Response · Authors · 2021-11-23
> **Please provide your constructive and technical review**
>
> Dear Reviewer,
>
> It's hard to find such information without using a special tool. In fact, you can just find our work at Arxiv. It's normal in ICLR community.
> Using such a reason to reject is unfair to our work.
>
> It's highly appreciated if you could further provide your constructive and technical review. Thanks in advance.

---

### Official Review · Reviewer_7Cc7 · 2021-11-03

**Correctness:** 3
**Technical Novelty And Significance:** 2
**Empirical Novelty And Significance:** 2
**Recommendation:** 3
**Confidence:** 4

**Main Review:**

Strengths:
+ The label deficiency problem is a realistic gap before we put federated learning into practice. The research question itself is a good and convincing topic to explore
+ They cite appropriate related research work including SSL and Personalized FL.
+ The problem formulation and the optimization function define the research problem clearly. They provide experiment results to support the claims. The analysis and discussion after the experiment are sufficient and convincing.

Weakness:
- I am a little bit concerned about the technical novelty in this paper. To combine SSL and FL is fine but lacks the motivation and we may not encourage just merging existing techniques together in a framework.
- Also, by adding regulations, adding constraints, or other hyper-parameters to control the difference between the global models and locals, such approaches have been proposed already. The hidden ideas or we say the core idea is quite similar.
- More discussion is expected about how to choose hyper-parameters.

**Summary Of The Paper:**

The authors merge self-supervised learning in personalized federated learning to solve the limited label and data heterogeneity problems in the local clients. They test several current algorithms under their framework. Then they propose an algorithm named Per- SSFL considering the balance between the consensus and personalization. Finally, they provide experiment results to support their claims and comprehensive analysis of what they find. The main contribution is to design a self-supervised FL framework with supportive experiment results. They also provide suggestions to choose appropriate algorithms and hyper-parameters under different settings.

**Summary Of The Review:**

In FL, label deficiency and data heterogeneity are two of the main open research questions. Authors propose a natural idea to leverage SSL in FL with the support of testing, analysis, and experiments. However, I am not sure the novelty of this work is good for this conference. I might expect more novel technique ideas in this paper-increasing the FL field.

---

> ### Author Response · Authors · 2021-11-23
> **Our work is novel! It is unfair to conclude that our method is just adding regularization.**
>
> We believe the following contributions build a solid baseline for research in this direction, and our open-source code may further enhance this endeavor.
>
> 1. (Global-SSFL) A self-supervised federated learning framework for a global model.
>
> 2. Global-SSFL’s interpretability in optimization perspective.
>
> 3. (Per-SSFL) A novel self-supervised federated learning framework for personalized models. The optimization method is novel, compared to the simple combination of Per-FL and SimSiam. Note that the optimizer we proposed is based on our large-scale experimental exploration, it's not that straightforward.
>
> 4. (Many algorithmic variants of Per-SSFL) We also proposed many variants of Per-SSFL that does not perform better than our novel formulation and algorithm. These algorithms serve as the baseline for our novel Per-SSFL.
>
> 5. A new evaluation metric for Per-SSFL. Please check Section 3.
>
> 6. A real distributed training system makes this research possible (evaluating different algorithmic variants in a reasonable time; exploring the memory constraint led by large batch size training on SSL, etc.). It’s also very important for future research in this area.
>
> In addition, we want to emphasize that *our core contribution is that SSFL is the first unified framework that tackles two core challenges in the ML aspect of federated learning: label deficiency at the edge and data distribution heterogeneity*. Until now, FL community doesn’t have such a learning framework. For the first challenge, we demonstrate that the standard FedAvg algorithm is compatible with recent breakthroughs in centralized self-supervised learning, such as SimSiam networks. This is not trivial since SimSiam-like framework normally requires larger batch size to work. Our careful design makes it works in FL setting for learning a global model.
>
> However, solely relying on such integrated design is insufficient towards a practical FL learning framework. As such, to address data heterogeneity at the edge devices in this framework simultaneously, we have innovated a series of algorithms that broaden existing supervised personalization algorithms into the setting of self-supervised learning, including perFedAvg, Ditto, and local fine-tuning, among others. We further propose a novel personalized federated self-supervised learning algorithm, Per-SSFL, which balances personalization and consensus by carefully regulating the distance between the local and global representations of data. These algorithmic variants are novel in ML literature, we empirically analyze and compare their performance and figure out the optimal design. In addition, we contributed an open source distributed training system for this kind of research. This is important because SimSiam-like framework is relatively larger models than other models (shallow CNNs, LSTM) used in current FL literature. We believe our algorithmic and system contribution can ignite the trend of research in handling two core challenges in a unified framework.

---

### Official Review · Reviewer_TsoV · 2021-11-03

**Correctness:** 3
**Technical Novelty And Significance:** 2
**Empirical Novelty And Significance:** 2
**Recommendation:** 5
**Confidence:** 3

**Main Review:**

Strength:

+ The self-supervised learning topic for FL is emerging and has great practical value.
+ Both non-personalized and personalized FL settings are investigated in the SSL regime.
+ Experiments and ablation studies are conducted under different settings.

Weakness:

- For the Per-SSFL framework, the local (personalized) model and global model are used. The memory consumption aspect should be discussed. For resource-constrained edge clients, high memory cost could be an issue.
- Although it mentioned in the implementation setting that the client number selected per round is 10, it is not clear how many total clients are used in the FL setting.
- In Table 1, what is the FL method under the supervised setting?
- In Figure 2, what does (0.5) for SSFL on IID and SSFL on non-IID mean?
- As can be seen in Figure 2, the convergence rates for the IID and non-IID cases are quite similar. Can you provide an explanation for that?
- After reading Appendix D and Figure 10 in Appendix, the experimental setting on GLD-23K is still not quite clear. For example, how the local training set for each client is generated? What about the label distribution? Also, it seems that the number of clients used on the GLD-23K is different from that on CIFAR-10.
- For ease of comparison and implementation, it would be good to evaluate the method on more commonly used datasets such as CIFAR-100 and Tiny-ImageNet and other datasets besides vision datasets (e.g., text) for FL.
- It would be interesting to see the SSFL results under different numbers of selected participant clients.
- Since $\lambda$ is an important parameter that balances consensus and personalization, its effect should be studied.

Minor issues:

1. Section 2.1, “…  the local empirical risk over the heterogeneous dataset $D^k$.” -> $D_k$
2. Figure 10 appeared in Sec. 5.2 and 5.3, it should be Fig. 2 and Fig. 3.
3. In Figure 2(a), the colors for the curve (SSFL on non-IID) and its legend are different (pink vs. red). It should be made consistent.
4. Sec. 3.2, “… contemporary self-supervised learning frameworks (e.g., SimSiam,
SwAV, BYOL)” -> should be “… (e.g., SimCLR, SwAV, BYOL)”

A careful proofread of the paper is highly recommended.


**Summary Of The Paper:**

This paper introduces the self-supervised learning (SSL) framework for FL. Different SSL methods are investigated to study their feasibility under the FL setting. With the popular SimSiam framework, personalized federated SSL is proposed. The performance comparison indicates
that representation regularization-based personalization method is able to outperform other variants.

**Summary Of The Review:**

I appreciate the emerging topic of SSL in FL. The proposed personalized federated SSL is interesting. However, there are quite a few unclear details regarding the method and experimental settings.

---

> ### Author Response · Authors · 2021-11-23
> **Author Response**
>
> * Q1: For the Per-SSFL framework, the local (personalized) model and global model are used. The memory consumption aspect should be discussed. For resource-constrained edge clients, high memory cost could be an issue.
>
> Response: Memory is not the concern in model interpolation-based FL, given that at any timestamp, only one optimizer is working. Thus, we don’t need to cache two times the memory size of the single model. We also reduce the memory cost by the experimental analysis in Section 5.4.1. As we can see, the most commonly used “gradient accumulation” can address the concern.
>
> * Q2: Although it is mentioned in the implementation setting that the client number selected per round is 10, it is not clear how many total clients are used in the FL setting.
>
> Response: For CIFAR-10, both the client per round and the number of the total clients are 10. For GLD-23K, The gld23k dataset contains 203 classes, 233 clients, and 23080 images. https://www.tensorflow.org/federated/api_docs/python/tff/simulation/datasets/gldv2/load_data
>
>
> * Q3: In Table 1, what is the FL method under the supervised setting?
>
> Response: It's naive FedAvg.
>
> * Q4: In Figure 2, what does (0.5) for SSFL on IID and SSFL on non-IID mean?
>
> Response: we describe alpha (0.5) at Section 5 - “Dataset”: For the non-I.I.D. setting, we distribute the dataset using a Dirichlet distribution (Hsu et al., 2019), which samples p∼Dir(alpha) and alpha = 0.5.
>
> * Q5: As can be seen in Figure 2, the convergence rates for the IID and non-IID cases are quite similar. Can you provide an explanation for that?
>
> Response: It’s not similar. The gap is 6%, also summarized in Table 1.
>
> * Q6: After reading Appendix D and Figure 10 in Appendix, the experimental setting on GLD-23K is still not quite clear. For example, how the local training set for each client is generated? What about the label distribution? Also, it seems that the number of clients used on the GLD-23K is different from that on CIFAR-10.
>
> Response: For GLD-23K, The gld23k dataset contains 203 classes, 233 clients, and 23080 images. Please check this at Google TFF: https://www.tensorflow.org/federated/api_docs/python/tff/simulation/datasets/gldv2/load_data
>
>
> Q7: For ease of comparison and implementation, it would be good to evaluate the method on more commonly used datasets such as CIFAR-100 and Tiny-ImageNet and other datasets besides vision datasets (e.g., text) for FL.
> It would be interesting to see the SSFL results under different numbers of selected participant clients.
>
> Response: For CIFAR-10, GLD-23K, we use different numbers of clients, so it should be fine.
>
> Q8: Since λ is an important parameter that balances consensus and personalization, its effect should be studied.
>
> Response: Thanks for your suggestions, we will study this in our revision.
>
> For other minor issues, we've revised accordingly. Thanks for your careful proofreading.

---

### Decision · Program_Chairs · 2022-01-20

**Decision:**

Reject

**Comment:**

This manuscript proposes an extension of semi-supervised learning to the federated setting. The contributions include a thorough evaluation of performance and some method extensions.

There are four reviewers. One reviewer points out a name leakage issue in the code that was missed and suggests deks-rejection. The area chair has chosen not to desk-reject the paper. Three other reviews agree that the manuscript addresses an interesting and timely issue -- indeed, label acquisition is a significant issue in federated learning. Three reviewers agree to reject the paper -- raising concerns about novelty compared to existing methods, some details of the evaluation, and some lack of clarity. The authors provide a good rebuttal addressing many of these issues. However, the reviewers are unconvinced that the method is sufficiently novel after reviews and discussion. Authors are encouraged to address the highlighted concerns for future submission of this work.